# COVID-19 Vaccine Acceptance among Liver Transplant Recipients

**DOI:** 10.3390/vaccines9111314

**Published:** 2021-11-11

**Authors:** Andrea Costantino, Federica Invernizzi, Erica Centorrino, Maurizio Vecchi, Pietro Lampertico, Maria Francesca Donato

**Affiliations:** 1Division of Gastroenterology and Endoscopy, Fondazione IRCCS Ca’ Granda Ospedale Maggiore Policlinico, 20122 Milan, Italy; maurizio.vecchi@unimi.it; 2Department of Pathophysiology and Transplantation, Università degli Studi di Milano, 20122 Milan, Italy; erica.centorrino@unimi.it; 3Division of Gastroenterology and Hepatology, Fondazione IRCCS Ca’ Granda Ospedale Maggiore Policlinico, 20122 Milan, Italy; federica.invernizzi@policlinico.mi.it (F.I.); pietro.lampertico@unimi.it (P.L.); francesca.donato@policlinico.mi.it (M.F.D.); 4CRC “A. M. and A. Migliavacca” Center for Liver Disease, Department of Pathophysiology and Transplantation, Università degli Studi di Milano, 20122 Milan, Italy

**Keywords:** COVID-19 vaccine, COVID-19, vaccination hesitancy, vaccine hesitancy, liver transplantation, liver transplant

## Abstract

(1) Background: COVID-19 vaccination hesitancy is a threat for fragile patients. We aimed to evaluate COVID-19 vaccination hesitancy and its reasons in a population of liver transplant (LT) recipients. (2) Methods: In February 2021, a questionnaire on COVID-19 vaccines was sent to LT patients followed at our liver transplant outpatient clinic in Milan, Italy. Sociodemographic and clinical characteristics were recorded. Patients were defined as willing, hesitant, or refusing and their reasons were investigated. Associations between baseline characteristics and willingness were evaluated. Since March 2021, when the COVID-19 vaccines became available for LT candidates and recipients in Italy, the entire cohort of LT recipients was contacted by phone and called for vaccination, and the rate of refusals recorded. (3) Results: The web-based survey was sent to 583 patients, of whom 190 responded (response rate of 32.6%). Among the respondents to the specific question about hesitancy (184), 157 (85.3%) were willing to be vaccinated against COVID-19, while 27 (14.7%) were hesitant. Among the hesitant, three were totally refusing, for a refusal rate of 1.6%. Thirteen hesitant patients (48.1%) answered that their COVID-19 vaccination hesitancy was influenced by being a transplant recipient. The fear of adverse effects was the main reason for refusal (81.5%). Of the 711 LT patients followed at our center, 668 got fully vaccinated, while 43 (6.1%) of them refused the scheduled vaccination. (4) Conclusions: Most patients accepted COVID-19 vaccines, although 6.1% refused the vaccine. Since it is crucial to achieve adequate vaccination of LT patients, it is very important to identify the reasons influencing COVID-19 vaccination hesitancy so that appropriate and targeted communication strategies can be established and specific vaccination campaigns further implemented.

## 1. Introduction

Severe acute respiratory syndrome coronavirus 2 (SARS-CoV-2) mRNA-based and viral vector-based vaccines are approved for the general population [1,2]. These vaccines provide effective protection against coronavirus disease 19 (COVID-19) [3,4]. Nevertheless, concerns about their efficacy and safety have been reported. The term “vaccination hesitancy” refers to the “delay in acceptance or refusal of vaccination despite availability” according to the World Health Organization [WHO] Strategic Advisory Group of Experts on Immunization [SAGE] Working Group on Vaccine Hesitancy [5]. Even before the COVID-19 pandemic, vaccination hesitancy was recognized by the WHO as being one of the 10 threats to global health, since vaccine uptake has been declining worldwide [6].

Solid organ transplant recipients are at increased risk of infection due to the chronic immunosuppressive therapy required to prevent graft rejection; in this setting, vaccinations are helpful to avoid many infectious diseases. The risk of infection is usually higher in the first six months after liver transplant due to the intensive immunosuppressive regimens required; for this reason, vaccinations are scheduled for people on the transplant waiting list. Live attenuated vaccines are contraindicated in this population, due to the risk of active vaccine-induced infection. Liver transplant (LT) recipients have multiple potential risk factors of poorer outcome in the case of SARS-CoV-2 infection because of their life-long immunosuppression and high prevalence of comorbidities. However, LT is not independently associated with death when matched to control patients with similar comorbidities. Age, renal function, neoplasms, and other risk factors associated with poor prognosis in patients with COVID-19 in the general population appear to be more important in determining outcome than LT [7].

At the beginning of the COVID-19 pandemic, LT recipients were strongly advised to follow all normal preventative measures, including social distancing and the correct use of masks. COVID-19 vaccination is also strongly recommended for LT recipients by the Italian (AISF) [8], European (EASL) [9], and American (AASLD) [10] liver associations. The aim of our study was to assess COVID-19 vaccination hesitancy among LT patients.

In Italy, the COVID-19 vaccination campaign began at the end of December 2020, when the first vaccines were administered to frontline healthcare workers and nursing home staff and residents. In February, the campaign was extended to the general public, targeting the priority groups of those over 80 and those working in key sectors, including schools, universities, prisons, and the Armed Forces. Since March, vaccination has been offered to those at very high risk of becoming severely ill with COVID-19, such as the liver transplant recipients, followed by the general population according to age.

COVID-19 vaccination is offered free of charge either at our transplant center or in the local vaccination center closest to the patient’s home.

## 2. Materials and Methods

### 2.1. Study Design

Between January and February 2021, a web-based anonymous questionnaire [Appendix A].was sent to LT patients followed at our liver transplant outpatient clinic in Milan, Italy. The questionnaire was adapted from a previous validated questionnaire evaluating vaccination hesitancy through the investigation of sociodemographic, disease-related, and lifestyle data, as well as attitude to vaccinations in general and to COVID-19 vaccines in particular [11].

The questionnaire was formulated online on the EUSurvey platform by our center. Patients were invited to complete the web-based questionnaire through a text message or phone call and then received an email containing the questionnaire URL—at the start of which the patients were required to provide their informed consent before filling in the questionnaire. Completion of the anonymous web-based survey did not result in any benefit or financial compensation for the patients. The patients were classified as willing, hesitant, or refusing (hesitant patients not intending to be vaccinated against COVID-19 at all) to accept COVID-19 vaccination, and the reasons and possible factors influencing vaccination hesitancy were investigated with multiple-choice questions.

Considering lifestyle questions, attitudes toward vaccines were investigated through the following question: “What kind of attitude do you think you have towards vaccinations in general?” Participation in prevention programs was self-reported with the question: “Do you undergo preventive diagnostic activities (e.g., pap-test, mammography, and fecal occult blood test)?” Active lifestyle was intended in the questionnaire as regular physical activity (e.g., individual sport, running, gym) [Appendix A].

When the COVID-19 vaccines became available for LT candidates and recipients in Italy (March 2021), the entire cohort of LT recipients was contacted by phone and called for vaccination, and the rate of refusals was recorded.

COVID-19 vaccines were offered independently of the questionnaire responses.

### 2.2. Statistical Analysis

Absolute and relative frequencies were calculated for the categorical (qualitative) variables, and quantitative variables were summarized by their means. All variables found to have a statistically significant association with vaccination hesitancy/refusal in the univariate analysis were included in a multivariate backward stepwise logistic regression model. The level of significance chosen for the multivariate logistic regression analysis was 0.05. Statistical analysis was performed using SPSS software 26.0 (IBM, Armonk, NY, USA), and statistical significance was set at *p* < 0.05.

## 3. Results

The web-based questionnaire was sent to 583 patients who were feasible to answer to a web-based survey among the 711 followed at our center, of whom 190 responded (response rate of 32.6%). Most respondents were men (68.4%), with a median age of 64 years. The great majority of the respondent LT recipients were adhering to their treatment (87.9%) and following prevention programs (78.9%) (Table 1).

Six patients (3.1%) did not answer the specific question on vaccination hesitancy: “Would you accept vaccination against COVID-19 tomorrow?”Among the respondents to this specific question (184), 157 (85.3%) were willing to be vaccinated against COVID-19, while 27 were not (14.7%). Among the hesitant, three were refusing, for a refusal rate of 1.6%. The reasons most frequently given by hesitant patients to justify their refusal (more than one could be indicated) was the fear of adverse events in 22/27 and concerns about the rapid development of COVID-19 vaccines in 17/27 (Figure 1).

Thirteen hesitant patients (48.1%) answered that their COVID-19 vaccination hesitancy was influenced by being a transplant recipient, while 92.3% (145) of patients willing to be vaccinated thought that their condition was a valid reason to prioritize vaccination. None of the possible determinants were significantly associated with vaccination hesitancy/refusal in univariate analysis, apart from a positive attitude toward vaccines in general (Table 2), not confirmed in the multivariate analysis.

Over the last six months, since the beginning of the vaccination campaign for LT patients (March 2021), we contacted by phone/mail all LT recipients followed in our center, both living in our and other Italian regions. Among our 711 contacted patients (mostly men (70.6%), with a median age of 63 years (19–84), [Appendix A], 668 got fully vaccinated with BNT162b2 of Pfizer/BioNTech, (New York, NY, USA) or with mRNA-1273, Moderna (Cambridge, MA, USA), while 43 (6.1%) of them refused the scheduled vaccination (based on our last check on 31 August).

## 4. Discussion

To the best of our knowledge, among LT recipients, this is one the first reports to investigate both intention to receive the COVID-19 vaccine (with its reasons and possible determinants) and real-life acceptance. This study was conducted in a liver transplantation center in Lombardy, the first region in Europe to be badly affected, so it is likely that the respondents were fully aware of the severity of COVID-19.

Among our LT patients who answered to the web-based survey, 14.7% were classified as hesitant, but ultimately, very few (1.6%) intended to totally refuse the vaccine. These rates were lower than those found in a contemporaneous global survey of the general populations in 19 different countries (8.1%) [12]. A national Italian survey of the general population (12,322 respondents) showed that only 65.2% of respondents intended to be vaccinated against COVID-19 as soon as a vaccine was available. Likewise, the refusal rate (totally opposing) of this general population group was higher (17.6%) compared to ours (1.6%) [13]. Nevertheless, the populations are not crudely comparable, and comparisons should be extrapolated with caution in the absence of matched analysis. The same questionnaire was sent to a cohort of patients with coeliac disease of the same geographical area (followed at our hospital) during the same period of time (February 2021). In the cohort of LT patients, we observed a lower percentage of hesitancy (14.7% vs. 25.2%) compared to the coeliac patients. However, the two groups were not matched by sex and age [14].

Nonetheless, the “real” refuse of COVID-19 vaccination in LT patients when called by phone and asked if they had gotten vaccinated was higher than in the survey (6.1% vs. 1.6%). This difference is likely to be explained by a response bias, because those who filled in the questionnaire (1/3 of the patients) may have had a more favorable attitude toward the vaccine than those who did not. This hesitancy rate is higher than in another recently published smaller cohort of Italian LT recipients (6.1% vs. 3.4%) [15]. As regards general vaccinations, 83.2% of our respondents had a positive attitude toward vaccines and they intended to be vaccinated again in the future. These findings are comparable to the results shown in a study on the general population, surveyed before the pandemic. In this earlier study, 12.7% of respondents were hesitant about vaccinations, while 4.7% declared they were completely against vaccinations [11]. LT recipients could be considered a population familiar with vaccinations, since they receive vaccines while on the transplant waiting list and are advised to be vaccinated annually against certain infective agents (e.g., flu). These patients also receive frequent medical controls to manage immunosuppression levels, associated complications, and comorbidities. These aspects could justify the high rate of adherence to prevention programs and the positive attitude toward COVID-19 vaccination shown in our survey.

This survey allowed us to better understand the major fears determining vaccination hesitancy/refusal to these new COVID-19 vaccines, which were uncertainties regarding safety and efficacy. In this respect, a recent controlled study from Israel showed that mRNA-based vaccines are well tolerated with no major adverse events in LT recipients compared to a control group, but that LT recipients had a significantly lower serological response against SARS-CoV-2 compared to controls (47.5% vs. 100%) [16]. The issue of immunogenicity should be evaluated further on T memory cells and not only on the serological response. Thus, further data on efficacy to two doses of COVID-19 vaccine among the LT recipients may help reducing vaccination hesitancy in some residual hesitant. Considering patients with liver cirrhosis, recently published efficacy clinical data in a population of 20,000 cirrhotic patients (LT recipients were excluded) showed that mRNA vaccine administration was associated with an excellent reduction in COVID-19-related hospitalization or death [17].

A plausible explanation for our high willingness rate could be motivated by the great prevalence of the Catholic religion in Italy, as the Catholic Church has endorsed the COVID-19 vaccination, influencing its acceptance [18,19]. The possible correlation of COVID-19 vaccination hesitancy and SARS-CoV-2 variants was not investigated in our questionnaire in February 2021. The delta variant had a prevalence ranging from 5.2% in May to 77.7% at the end of July 2021 [20]. The vaccination hesitancy in our LT patients was not correlated with immunosuppressive drugs, as observed in a large national cohort of patients with inflammatory bowel diseases [21]. As a limitation of our study, we did not investigate whether patients’ residence distance from the vaccine center might have influenced vaccination hesitancy. However, the COVID-19 vaccine was offered at our transplant center or in the local vaccination center closest to the patient’s home.

Attitudes toward vaccination can be seen as a continuum, ranging from total acceptance to complete refusal, a situation which is complex and context-specific, varying in different countries. Among the vaccine-hesitant patients, those refusing vaccination are the most difficult to convince [22]. Factors such as complacency, convenience, and lack of confidence in vaccines may all contribute to vaccination delay or refusal of one, some, or most vaccines [22]. Larson et al. highlighted the fact that the vaccine community has been negligent in demanding rigorous research to understand the psychological, social, and political factors that affect public trust in vaccines [23]. The SAGE Immunization Working Group on Vaccine Hesitancy proposed a set of recommendations directed at the public health community and WHO Member States, which highlights the importance of focusing on the need to increase understanding of vaccination hesitancy and its determinants [24].

## 5. Conclusions

In conclusion, since it is crucial to achieve adequate COVID-19 vaccination of LT patients, it is important to identify the reasons influencing hesitancy so that appropriate patient–doctor communication can be established and specific vaccinations campaigns further implemented in all transplant recipients. To date, hesitancy of the third dose of the COVID-19 vaccine for at-risk patients is the next issue to address. As recently indicated by the Permanent Transplant Commission of the AISF [25], homogeneous international recommendations for LT recipients are required in order to plan the continuation of an effective and rapid vaccination campaign.

## Figures and Tables

**Figure 1 vaccines-09-01314-f001:**
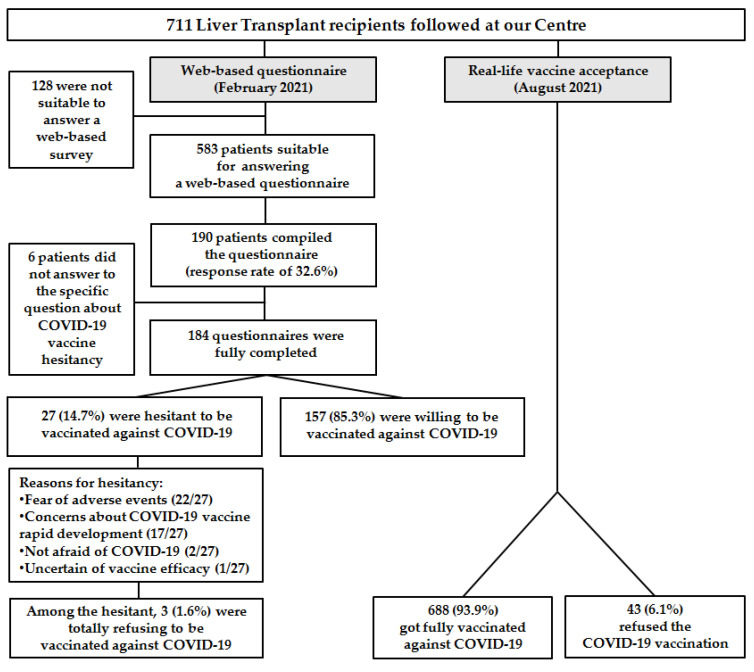
Flow diagram of the study of our liver transplant population showing, on the left side, COVID-19 vaccination willingness and hesitancy (web-based questionnaire) and, on the right side, the real-life COVID-19 acceptance.

**Table 1 vaccines-09-01314-t001:** Sociodemographic, lifestyle, and clinical characteristics of liver transplant recipients who answered the questionnaire.

Characteristic	Overall * (n = 190)
Age, years, median (range)	63 (23–81)
Male gender, n (%)	130 (68.4%)
Educational level, n (%)	
Undergraduate	161 (84.7%)
Graduate	29 (15.3%)
Years from transplant, n (%)	
<5 years	62 (32.6%)
5–10 years	51 (26.8%)
>10 years	77 (41.1%)
Transplant indication, n (%)	
Cirrhosis	104 (54.7%)
Hepatocellular Carcinoma	46 (24.2%)
Others	40 (21.1%)
Immunosuppressive therapy *	
Tacrolimus	167 (87.9%)
Cyclosporine	17 (8.9%)
Mycophenolate	98 (51.6%)
Azathioprine	5 (2.63%)
Everolimus	18 (9.5%)
Adherence to therapy, n (%)	
Yes	167 (87.9%)
No/not at all	23 (12.1%)
Social alcohol intake (≤2 standard drinks), n (%)	27 (14.2%)
Active lifestyle, n (%)	88 (46.3%)
Prevention programs, n (%)	150 (78.9%)
Positive attitudes towards vaccines, n (%)	158 (83.2%)

Note. * More than one could be indicated.

**Table 2 vaccines-09-01314-t002:** Sociodemographic, lifestyle, and clinical characteristics of patients willing or hesitant to receive the COVID-19 vaccine.

Characteristic	Overall * (n = 190)	Willing (n = 157)	Hesitant (n = 27)	*p*-Value
Male gender, n (%)	130 (68.4%)	110 (70.1%)	15 (55.6%)	0.179
Median age, years (range)	63 (23–81)	64 (23–81)	58 (25–74)	0.211
Therapy adherence, n (%)	167 (87.9%)	136 (86.6%)	26 (96.3%)	0.207
Active lifestyle, n (%)	88 (46.3%)	71 (48.2%)	14 (51.9%)	0.538
Prevention programs, n (%)	150 (78.9%)	124 (79%)	22 (81.5%)	1.00
Positive attitude toward vaccination, n (%)	158 (83.2%)	147 (93.6%)	8 (29.6%)	0.0001

Note. * Six patients (3.1%) did not answer the question: “Would you accept vaccination against COVID-19 tomorrow?”.

## Data Availability

The datasets generated and/or analyzed during the current study are not publicly available but are available from the corresponding author upon reasonable request.

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
