# Peer review of "COVID-19 Vaccine Acceptance among Liver Transplant Recipients"

_vaccines, 2021, doi:10.3390/vaccines9111314_

Round 1

Reviewer 1 Report

Thank you for conducting this survey and for submitting your data for publication.

Please include in the introduction data on the epidemiology of the Delta variant in Italy during the months of February through August 2021, since this may have influenced responses.

Please include a copy of the questionnaire in the Methods section.

I commend you for allowing respondents to this survey to remain anonymous, since this may limit survey uptake and influence the responses. I also commend you for not attaching survey completion with any benefit or financial compensation.  

Please include in the Methods section definitions for “vaccine educated”, “therapy adherence”, “active lifestyle”, “prevention programs” and “positive attitude toward vaccines.”

Please include in the Methods section whether the reason(s) for hesitancy were surveyed as an open-ended question, or multiple choice question.

Please state whether the COVID-19 vaccine was offered at cost or free of charge, and location where it was offered; at the transplant center or local pharmacies and local physician offices.

Please clarify the denominator for your cohort; 583, or 711? I suspect the 583 are among the 711, but anonymous responses would not identify respondents.

I recommend a flow diagram for your study showing total cohort, survey respondents and non-respondents, and responses of respondents.

I recommend elimination of data for the 6 patients who did not respond to the main question of the survey conducted in February 2021.

Please include an online only table for the whole cohort showing basic demographic data in order to put in perspective the characteristics shown in table 1.

Please include data on interval between transplant and survey deployment for both survey respondents and non-respondents.

Please include data on patients’ residence distance from the transplant center, or wherever the COVID-19 vaccine was offered for both survey respondents and non-respondents.

Page 5, line 174: I think you meant “immunogenicity”, not “efficacy”.

Author Response

We thank the reviewer for the precise comments. 

a) we included in the discussion data on the epidemiology of Delta variant, but impact of delta variant on hesitancy was not investigated in the questionnaire sent in February

b) we included the questionnaire in the Supplementary File

c) we emphasized in the section Materials and Methods (Study Design) that the questionnaire was anonymous and did not give any benefit to the patient

d) we better defined what we intended in the questionnaire for therapy adherence”, “active lifestyle”, “prevention programs” and “positive attitude toward vaccines. (Note of Table 1)

e) reasons for hesitancy were investigated in a multiple choice question, we add a sentence in the section Materials and Methods (Study Design)

f) we stated that COVID-19 vaccine was offered free of charge and where it was offered (Introduction)

g&h) we added a flow diagram of the study showing total cohort, the survey respondents and non-respondents and responses of respondents, in this way we eliminated the whole previous study. 190 patients answered among the 583 patients who were feasible to answer to a web-based survey. We better clarified it in the results. 

i) we eliminated the data of 6 patients and recalculated the percentages (Abstract, Results, Discussion)

l) we added a table showing the characteristics of the whole cohort (Table 1, Results)

m&n) Data of transplant is in Table 1 (Results), we have not had the possibility to answer to question of distance, since it was not present in the survey. Indeed, we added that the vaccine was offered for both survey respondents and non-respondents Materials and Methods (Study Design)

o) Thanks, we meant immunogenicity (Discussion)

Reviewer 2 Report

The authors report on the Covid vaccination attitudes among liver transplant recipients in a northern Italian population.   Ultimately , 94% of this population was successfully vaccinated which is an exceptionally positive outcome.  The survey was sent out in January/ February of 2021 during the early phases of the vaccine role outs. Most survey respondents had either a favorable or hesitant attitude toward Covid vaccination.  Only 1.5 % responded that they would refuse vaccination.   The survey was not able to detect significant difference in social or gender characteristics between the attitudes of the respondents.    The paper would be much stronger if differences were identified but ultimately the small number of those who refused may have severely limited the power to detect those differences.   One critical issue which was  not addressed by the authors was religious belief.  Approximately 77% of Italians identify as Catholic and the Catholic church has endorsed Covid vaccination.   In the United States, many who are opposed to vaccination are seeking religious exemptions.  It would be interesting to know if religious views affected the opinions of Covid vaccination in this Italian population.

Author Response

We thank the reviewer for the estimation of our work. We also thank the reviewer for her/his interesting comment. 

Yes, we do agree that the high percentage of Italians who are likely to be Catholic may have influenced the low COVID-19 vaccine hesitancy rate. However, among the patients who answered the questionnaire, their religion was not investigated with a specific question. 

However, we do agree that it is likely that the high willingness rate among our population could be attributable to the Italian Catholic faith because most of our patients are Italian (about 90%), and other studies have shown this higher vaccination acceptance rate among Catholics compared to other faiths.

We added a small paragraph in discussion section. 

Reviewer 3 Report

Proposed paper is interesting and well written. I have only one point to be defined before paper can be accepeted for pubblication. Are data on clinical LT characteristics available, such as an example, indication to transplantation, disease duration before transplantation, years from transplantation, type of immunosuppresant? In fact it could be possible that such factors are part of the hesitancy (i.e. being immunosuppressed with a more strong drug give the patients the idea that the vaccines could be dangerous, being aftected of colangiocharcinoma and mybe for long time give the patients the feeling of being moe fragile, and so on.

If you can complete with this information and maybe try to see if they correlate with vaccines hesitancy it could enhance your paper. If those information are not available please comment on this issue in the discussion.

Author Response

We thank the reviewer for the estimation of our work. We also thank the reviewer for her/his interesting comment. 

Yes, data on some clinical characteristics of LT patients were available. In the anaonymous questionnaire patients had to answer to questions related to their disease.

We added a table (Table 1) with the characteristics of patients who answered the questionnaire (these questions included indication fo liver transplantation as liver cirrhosis or hepatocellular carcinoma, years from transplantation, type of immunosuppressant). None of these factors resulted, indeed, associated to vaccine hesitancy.  

Disease duration before transplantation was not a question of the survey. In our Center, patients with cholangiocarcinoma have not been placed on the LT waiting list, as it happens in other centers, also in our City. 

Round 2

Reviewer 1 Report

Thank you for revising your manuscript. 

Titles influence readers’ impression before reading the paper. For example, reference 15 explicitly states “High acceptance rate.” The title of your manuscript is in the form of a question, but it biases the readers to think “hesitancy” before reading the results. The difference in hesitancy rate between reference 15 and your study is small, and likely represents expected variation. I recommend a more moderate title for your manuscript, such as “Acceptance of liver transplant recipients to the first two doses of COVID-19 vaccine.”

I recommend you move definitions for “vaccine educated”, “therapy adherence”, “active lifestyle”, “prevention programs” and “positive attitude toward vaccines” from table 1 in the Results section to the Methods section.

What I meant by the whole cohort was the 711 patients in your program. Table 1 currently shows demographics for those who answered the questionnaire. Please include in a supplementary online table only basic characteristics of the 711 patients in order to put in perspective characteristics of patients who responded to the questionnaire shown in table 1. 

Please include data on interval between transplant and survey deployment for both survey respondents and non-respondents.

Please include in your discussion the limitation resulting from absence of data on patients’ residence distance from the transplant center, or wherever the COVID-19 vaccine was offered for both survey respondents and non-respondents. 

Please include in your discussion the limitation resulting from the fact that only a third of patients responded; limiting the generalizability of the outcome.

Author Response

We thank  the reviewer for her/his comments.

a) we changed the title to an affirmative one.

b) we moved definitions from Table 1 to Methods

c) we added a short table in the supplementary material with the most influent baseline socio-demographic characteristics of the total cohort of LT patients

d) unfortunately we do not have this data and we cannot answer this specific question

e&f) we added these limitations in the discussion

Reviewer 2 Report

Thank you for providing these updates.  It has provided additional clarity and I have no additional concerns. 

Author Response

Thank You